# Uropathogenic *Escherichia coli* Associated with Risk of Urosepsis—Genetic, Proteomic, and Metabolomic Studies

**DOI:** 10.3390/ijms26125681

**Published:** 2025-06-13

**Authors:** Beata Krawczyk, Paweł Wityk, Magdalena Burzyńska, Tomasz Majchrzak, Michał Jan Markuszewski

**Affiliations:** 1Department of Biotechnology and Microbiology, Faculty of Chemistry, Gdańsk University of Technology, 80-233 Gdańsk, Poland; magda.fordon@gmail.com; 2Department of Biopharmaceutics and Pharmacodynamics, Medical University of Gdańsk, 80-210 Gdańsk, Poland; pawel.wityk@gumed.edu.pl (P.W.); markusz@gumed.edu.pl (M.J.M.); 3Department of Analytical Chemistry, Faculty of Chemistry, Gdańsk University of Technology, 80-233 Gdańsk, Poland; tomasz.majchrzak@pg.edu.pl

**Keywords:** *Escherichia coli*, UPEC, virulence, sepsis, pathogenesis, OMICS technology, markers

## Abstract

In the absence of fully effective therapies and preventive strategies against the development of urosepsis, a deeper understanding of the virulence mechanisms of Uropathogenic *Escherichia coli* (UPEC) strains is needed. UPEC strains employ a wide range of virulence factors (VFs) to persist in the urinary tract and bloodstream. UPEC strains were isolated from patients with sepsis and a control group without sepsis. PCR was used to detect 36 genes encoding various groups of virulence and fitness factors. Profiling of both intracellular and extracellular bacterial proteins was also included in our approach. Bacterial metabolites were identified and quantified using GC-MS and LC-MS techniques. The UpaG autotransporter, a trimeric *E. coli* AT adhesin, was significantly more prevalent in urosepsis strains (*p* = 0.00001). Iron uptake via aerobactin and the Iha protein also appeared to be predictive of urosepsis (*p* = 0.03 and *p* = 0.002, respectively). While some studies suggest an association between S fimbriae and the risk of urosepsis, we observed no such correlation (*p* = 0.0001). Proteomic and metabolomic analyses indicated that elevated levels of bacterial citrate, malate, coenzyme Q10, pectinesterase (YbhC), and glutamate transport proteins, as well as the regulators PhoP two-component system, CpxR two-component system, Nitrate/nitrite response regulator protein NarL, and the Ferrienterobactin receptor FepA, may play a role in sepsis. These genetic biomarkers, proteins, and metabolites derived from UPEC could potentially serve as indicators for assessing the risk of developing sepsis.

## 1. Introduction

Based on the Global Burden of Disease study, the number of urinary tract infections (UTIs) increased by 66.45% between 1990 and 2021, affecting 4.49 billion people, with an age-standardized incidence rate (ASIR) of 5531.88 per 100,000 [1]. A 2.4-fold increase in deaths from UTIs was also observed between 1990 and 2019 [2].

*E. coli* is known as a commensal in the gastrointestinal tract but has also been reported as having strong pathogenicity due to different virulence factors (VFs) that facilitate infection of the urinary tract, and even invasion and dissemination to the bloodstream [3,4,5].

The important VFs of ExPEC strains include fimbriae types I, III, S, P, CNF1, and Dr; afimbrial adhesins (e.g., TosA, Afa); invasins (e.g., Ibe); toxins (e.g., Usp, HlyA, CNf-1); autotransporters (e.g., Adhesin Involved in Diffuse Adherence (AIDA-I) group, Serine Protease Autotransporters of Enterobacteriaceae (SPATE) group, Trimeric Autotransporter Adhesins (TAA) group); capsules; and iron uptake systems (e.g., siderophores—enterobactin, aerobactin, salmochelin, yersiniabactin).

The *E. coli* responsible for UTIs belong to the uropathogenic group (UPEC) with a specific set of virulence genes. The majority of UPEC virulence genes are found on pathogenicity islands (PAIs), where many genes are transferred from other species [6]. UPEC strains may exploit these for survival in the urinary tract of the human host.

According to Clermont et al. [7,8], *E. coli* strains belong to seven sensu stricto phylogenetic groups—A, B1, B2, C, D, E, and F—with the eighth group being the cryptic *Escherichia* clade I. These phylogenetic groups are characterized by a different set of VFs and fitness genes. Urovirulence was most commonly observed in the B2, F, and D groups [8,9,10,11].

Fimbriae are major protein structures contributing to the virulence of UPEC, including attaching to surfaces, forming biofilms, and evading the host immune response [12,13,14]. Type I fimbriae mainly initiate colonization of the bladder and play an important role in inducing the formation of bonds between neighboring cells and biofilm formations, in the central part of the tubule at a late stage of infection [15]. Type II fimbriae (P-pili) mediate attachment to the urethral mucosa and provide persistence in lower UTIs and pyelonephritis. These are more frequently reported in strains isolated from patients with urosepsis than in patients with cystitis [16]. S fimbrial adhesins bind to receptors on erythrocytes and epithelial cells of the renal tubules [17,18]. S fimbriae were detected in upper UTIs with renal failure, sepsis, and meningitis [19,20,21]. The type III fimbriae aids in the adhesion to endothelial and epithelial cells of the urinary bladder and kidneys, contributes to biofilm formation, and has the ability to agglutinate erythrocytes [22,23].

F1C-fimbrial adhesins were detected in strains for all types of UTIs and mediate binding to kidney tubular cells (renal failure) [24]. The Dr family of adhesins (Dr fimbrie, Afa afimbrial adhesin) play a very important role in adherence, invasion, and colonization of the urinary tract [25]. They induce polymerization of α-actinin and inhibit the process of exfoliation of bladder epithelial cells [20].

Attention has also focused on iron metabolism, participation in the oxidation-reduction reactions, and the phenomenon of ferroptosis due to catalysis of reactive oxygen forms and the consequence of injury and cell death [26]. Siderophores are considered VFs due to their ability to acquire iron from the host [27]. Enterobactin is produced by almost all strains, while aerobactin, salmochelin, and yersinabactin are found mainly in pathogenic strains [26,27,28,29,30].

For complicated UTIs, genes toxins *usp*, *hlyA*, and cytotoxic necrotizing factor-1 (*cnf1*) are often detected. Usp genotoxin, active against mammalian cells, is associated with strains associated with pyelonephritis and bacteremia [31]. Hemolysin HlyA is a type of pore-forming toxin that contributes to the development of cystitis, pyelonephritis, and sepsis by destroying immune cells such as macrophages in addition to urothelial cells [32]. CNF1 rearranges the actin filaments cytoskeleton, enhances the bacterial invasion, and increases the survival of uropathogenic bacteria within the urinary tract and blood [33,34,35].

Autotransporter (AT) proteins are among the major families of proteins secreted by Gram-negative bacteria, enabling them to colonize host tissues and establish infections [36,37]. The family of AT proteins secreted by *E. coli* strains is divided into three main groups: SPATE, TAA, and AIDA-I-type AT proteins [38]. AT proteins, which belong to the type V secretory pathway, can perform a variety of functions, including transport of molecules across the outer membrane, adhesion, serum resistance, hemagglutination, protease activity, biofilm formation, and toxin activity [39].

Buckles et al. (2009) used isogenic mutants and genetic complementation to prove that capsule expression may play an important role in UTIs caused by *E. coli* [40]. Capsules provide resistance to phagocytosis, protects against complement-mediated killing, and are very important in the pathogenesis of UTIs.

LC-MS–based proteomics enables the identification and quantification of proteins with high sensitivity and specificity, providing insights into bacterial virulence mechanisms, host–pathogen interactions, and antimicrobial resistance patterns. In the context of UTIs and urosepsis, LC-MS has been utilized to analyze bacterial outer membrane proteins, secreted VFs, and host immune response proteins. Host proteins involved in inflammatory pathways, such as cytokines (IL-6, IL-8, TNF-α) and complement system components (C3, C5a), have been detected at altered levels in response to *E. coli* infections. These host proteins, in combination with bacterial VFs, can serve as valuable biomarkers for distinguishing severe infections and guiding targeted therapeutic interventions.

In our studies, we looked at the differences in the composition of bacterial VFs and proteins/metabolites between uropathogenic strains and strains that caused urosepsis. We tried to answer the question of whether, apart from host factors, there are UPEC subtypes that are more predisposed to induce sepsis.

## 2. Results

### 2.1. Distribution of Virulence Factors in the Control Group vs. the Urosepsis Group Isolates

UPEC strains capable of causing urosepsis belong to a heterogeneous group of bacteria that exhibit varying degrees of virulence. We compared VFs among isolates from patients with urosepsis and those from patients with urinary tract infections (UTIs) without signs of sepsis.

The prevalence of VFs divided into specific groups is presented in Table 1 and Appendix A.

Statistically significant differences (*p* ≤ 0.05) were observed for virulence factors belonging to the autotransporter groups (*tosB*: *p* = 0.002; *aidA*: *p* = 0.003; *ag43*: *p* = 0.014; *upaG*: *p* = 0.00001: *upaH*: *p* = 0.033; *vat*: *p* = 0.0161, and *sat*: *p* = 0.0077); toxins (*usp* and *hly* (*p* = 0.0478 and *p* = 0.0385, respectively, but their prevalence was relatively low); and the iron uptake system, with aerobactin (*p* = 0.030 for *iuc* and *p* = 0.028 for *iut*) and *iha* (*p* = 0.002). Among the fimbrial adhesins, only the *sfa* gene encoding S fimbriae showed a statistically significant difference between groups (*p* = 0.0001). In our study, S-type fimbriae were significantly detected in the control group (63%; patients with UTI but without sepsis), whereas in blood isolates, S-type fimbriae were detected only in 31%. In our opinion, this is a factor that differentiates both research groups.

Due to the co-occurrence of VFs, the strains were grouped into three main clades for the urosepsis group and five main clades for the control group (UTI) (Appendix A). This indicates a greater genetic diversity of UPEC strains for the control group and more homogeneous virulence profile for urosepsis that may reflect common pathogenic mechanisms associated with bloodstream infections.

Autotransporters appear to constitute an important class of virulence factors. The distribution and coexistence of genes encoding components involved in the recognition and function of autotransporters in both the control and urosepsis groups are presented in Appendix A. The most common combinations were *upaG* + *aidA* and *upaG* + *aidA* + *ag43*, both of which were observed in 9.4% of urosepsis strains, as well as *upaG* + *aidA* + *ag43* + SPATE + *hbp*. Interestingly, the *boa* gene, which belongs to the SPATE family, was exclusively detected in urosepsis isolates and co-occurred with *pssA* in almost 5% of cases. By contrast, the control group exhibited the highest frequency of *upaG* + *aidA* (11.8%), with complex combinations of seven AT genes—including SPATE, *upaG*, *ag43*, *vat*, *tosB*, *pic* and either *upaH* or *aidA*—appearing less frequently. To evaluate patterns of gene co-occurrence within the autotransporter (AT) system, hierarchical clustering analysis (HCA) was also performed separately for the control group and the urosepsis group, using Jaccard distance and Ward linkage (Appendix A). In the control group, two major gene clusters were identified. The first cluster comprised *upaG*, *aidA*, *tosB*, *pic*, *vat*, *agn43*, and SPATE, indicating frequent co-occurrence of these genes in strains with lower virulence potential. A second, smaller cluster included *upaH* and *sat*, while genes such as *pssA*, *hbp*, and *pic-like* appeared more distantly grouped, suggesting limited presence or distinct functional roles. In contrast, the urosepsis group exhibited a more complex clustering pattern. A notably tight cluster involved *boa* and *pssA*—with *boa* being exclusively detected in urosepsis strains—suggesting its potential role in pathogenicity. Another extended cluster grouped *aidA*, *hbp*, *pic*, *tosB*, *upaH*, and *vat*, indicating coordinated expression of virulence-associated genes. Additionally, *upaG*, *agn43*, and SPATE formed a separate cluster, while *sat* and *pic-like* were grouped independently.

### 2.2. Binary Classification Results for Urosepsis Diagnostics

The results obtained for the urosepsis and control groups were juxtaposed and used for the binary classification based on the decision tree. The summary of this classifier is presented in Figure 1.

Most of the VFs are similarly distributed across both groups; however, it was possible to select two genes, *sfa* and *upaG*, that significantly differentiates the urosepsis and control groups. As presented in Figure 1A, the *sfa* is more frequent in the control group (63.5% in control group vs. 31.3% in the urosepsis group), whereas *upaG* is more common in the urosepsis group (93.8% vs. 67.1%). Their discrimination power was confirmed as those factors resulted in a much higher information gain than the other VFs. Thus, it was possible to build a decision tree classification model as shown in Figure 1C. The performance of this model was validated, and a total classification accuracy of 79.2% was obtained. The confusion matrix is depicted in Figure 1D and shows a relatively high true positive score of 76.6% and a high true negative of 81.2%.

### 2.3. Phylogenetic Groups and VFs Concerning Urosepsis vs. Control Groups

The *E. coli* isolates were classified into one of eight phylogenetic groups (A, B1, B2, C, D, F, and I [U]) using the Clermont scheme [7,8]. For both groups of isolates, no differences in phylogenetic group membership were observed. The most commonly detected phylogenetic group was B2, which is considered highly pathogenic. It was equally frequent in the control group and patients with urosepsis. The binary heatmap for the detection of VFs in each phylogenetic group is shown in Figure 2.

The combined data for the urosepsis and control groups were provided. As can be seen, there are certain patterns in the appearance of virulence genes in the phylogenetic group. The most variable ATs were confirmed in the B2 phylogroup with dominant *upaG* and *ag43* for the urosepsis group. The *chuA* is absent for A, B1, and C phylogenetic groups, irrespective of study groups. The *aidA* gene is less frequent for the B2 phylogenetic group and *ag43* for A, B1, and C; SPATE is dominant in the B2 phylogenetic group and seldom in other groups. Moreover, the *sfa*, *papG*, *kspMTII*, and *iha* factors are more frequent in B2. Type S fimbriae (*sfa* gene) in the urosepsis group, with few exceptions, occurred almost exclusively in group B2 and significantly less frequently than in the control group (*p* = 0.0001).

### 2.4. Proteomic and Metabolomic Profiling of Urosepsis Strains

Peptides uniquely identified in clinical *E. coli* strains from urosepsis by shotgun proteomics, including associated confidence scores, retention times, and theoretical and observed mass-to-charge (*m*/*z*) ratios, are of groups were observed. Figure 3 presents exclusively detected peptides for the urosepsis group that are not present in the strains in the control group.

A comparison of metabolites in clinical *E. coli* isolates from patients with urinary tract infection (UTI) and urosepsis is included in Appendix A.

#### 2.4.1. Citrate

In our proteomic study, citrate level was statistical significantly between *E. coli* isolated with urosepsis and the control group (*p* = 0.0224) (Figure 4).

#### 2.4.2. (S) Malate

The level of malic acid was higher for urosepsis isolates than the control group, and the difference was statistically significant (*p* = 0.027) (Figure 5).

#### 2.4.3. Ubiquinone (Coenzyme Q10)

The significant elevation of ubiquinone (Coenzyme Q10) in urosepsis samples (Figure 6) suggests an enhanced metabolic activity of *E. coli*, particularly in the context of biofilm formation and adhesion.

## 3. Discussion

### 3.1. Comparison of the Virulence of Strains Isolated from Urosepsis and the Control Group

Most *E. coli* strains isolated from the blood of patients hospitalized with sepsis originate from the upper urinary tract, and systemic infections caused by bacteria from the urinary tract are known as urosepsis [44]. Among the many VFs carried by uropathogenic *E. coli*, the most important are fimbrial and afimbrial adhesins, iron uptake systems (e.g., siderophores), toxins, and a group of proteins called ATs that can play an important role in the pathogenesis of urosepsis [45].

Fimbriae are major protein structures contributing to the virulence of UPEC [12,14]. In our study, type I fimbriae were found in more than 90% of strains, whether they caused urosepsis or not. Their function is critical for the first stage of urinary tract colonization. Floyd et al. (2015) showed a reduction in type I fimbriae expression under reduced oxygen conditions [46]. Also, Greene et al. (2015) proved that the function and expression of type I fimbriae in UPEC are decreased in human urine [47]. We can speculate that a decrease in oxygen may favor sepsis. The bacteria do not stay on the surface of the urinary tract, but spread to the kidneys and bloodstream, causing sepsis.

Fimbriae P was also frequently detected in UTIs and in sepsis (33% vs. 48%) but was not statistically significant. Among the control group, there were also patients with pyelonephritis, but sepsis did not occur in them; hence, we concluded that this is not a factor that has a direct impact on the development of sepsis. Nevertheless, it plays a role in the colonization of the upper urinary tract.

The affinity of S fimbriae for the extracellular matrix protein laminin may underlie the association of S fimbriae with sepsis [48,49]. In our study, quite surprisingly, S-type fimbriae were significantly detected in the control group. In our opinion, this is a factor that differentiates both research groups. The result was statistically significant (*p* = 0.0001). The remaining fimbriae—type III fimbriae and F1C and adhesins Afa and TosA—were less numerous and not statistically significant.

The frequency of the Usp and HlyA toxin genes was relatively low. Previous studies have suggested that certain strains downregulate the expression of the *hlyA* gene in the bloodstream to avoid triggering inflammation [50].

Depending on the type of AT, they exhibit functions such as adhesion, serum resistance, hemagglutination, protease activity, biofilm formation, and toxin activity [36]. In one-dimensional analysis, the *upaG* gene was the most common autotransporter (AT) in urosepsis isolates and statistically significant (*p* = 0.00001). UpaG, a TAA family member, promotes cell aggregation, biofilm formation, and adhesion to host tissues [36]. Previous research by Totsika (2012) has found that *upaG* are highly prevalent among pathogenic *E. coli* strains of intestinal as well as extraintestinal origins (92% of STEC strains and 86.5% of UPEC strains) [51]. Other ATs, such as Ag43 and UpaH, belong to the AIDA-I family and also contribute to virulence. The ability of a bacterium to form a biofilm is closely associated with its survival in the urinary tract environment [52,53], and the Ag43 protein has been detected in bacterial biofilms of the bladder epithelium of patients with chronic UTIs. Several variants of the Ag43 protein have been identified from different *E. coli* strains, showing significant sequence variability resulting in different aggregation and biofilm phenotypes [37,54]. UpaH is the largest AT-type AIDA-I protein and is abundant in UPEC strains causing sepsis. It has been implicated in bladder colonization and biofilm formation [55]. Unlike the other AT proteins, UpaH does not promote bacterial aggregation, adherence to bladder epithelial cells, or binding to extracellular matrix proteins. Nevertheless, expression of the UpaH strongly influences biofilm formation [37]. The secreted autotransporter toxin (SAT) belonging to the SPATE group exhibits cytopathic effects, causing vacuolization of bladder and kidney cells, among others. SAT also induces a host immune response, and has casein, factor V, and spectrin cleavage properties. Studies in a mouse model have also shown that the SAT protein plays a crucial role in UTIs, contributing to kidney damage [56]. Vacuolating autotransporter toxin (VAT), similar to SAT, induces vacuolization of eukaryotic cells. It was often found in avian pathogenic *E. coli* (APEC), but it was also identified in 65% of cystitis cases [57]. For urosepsis isolates, the *sat* gene occurred more often (*p* = 0.0077), but the second AT gene *vat* (*p* = 0.0161) was found more common in the control group. These ATs do not appear very often, but their presence is associated with cell destruction, which allows their access to deeper host tissues. Analysis of autotransporter (AT) gene co-occurrence (Appendix A) revealed distinct patterns, with combinations including SPATE, *ag43*, and *upaG* more common in urosepsis isolates, suggesting their association with increased virulence. The *boa* gene, exclusive to urosepsis strains and often paired with *pssA*, may play a key pathogenic role. Hierarchical clustering (HCA) (Appendix A) showed simpler gene groupings in control strains—where *upaG* clustered with lower-virulence genes—while urosepsis strains exhibited more complex clusters, indicating coordinated expression of virulence factors.

Our analysis revealed multiple siderophores, with aerobactin significantly enriched in urosepsis strains (*p* = 0.003), often co-occurring with enterobactin and yersiniabactin. Beyond iron acquisition, aerobactin enhances biofilm formation, oxidative stress resistance, and virulence [58,59]. Li et al. (2023), based on an analysis of human genetic signatures related to iron metabolism in sepsis using the GEO database, identified 21 sepsis signatures associated with iron metabolism [60]. There is likely to be a link between iron metabolic activity in bacteria and host iron metabolism as a risk for sepsis. Additionally, the *iha* gene, more prevalent in urosepsis (*p* = 0.002), encodes a membrane protein involved in iron uptake and is associated with recurrent UTIs and pyelonephritis [61,62]. These findings suggest a potential link between bacterial and host iron metabolism in sepsis and highlight IhaA as a possible marker for urosepsis risk.

Receptors for siderophores, IreA and IutA, demonstrated the ability to induce an immune response, both systemically and locally. However, the *fyuA* antigen provided an effective and long-lasting humoral response protecting against UPEC infections. A similarly important role in preventing UTIs was played by the salmochelin receptor IroN [63].

### 3.2. Proteomic and Metabolomic Studies

Identifying and validating specific bacterial proteomic and metabolomic biomarkers is promising for improving diagnostic accuracy and prognostic assessment in urosepsis. The results of the proteomics and metabolomics studies revealed increased levels of bacterial citrate, malate, CoQ10, pectin esterase, glutamate transport proteins, and the regulators PhoP two-component system (PhoP), CpxR two-component system (CpxR), Nitrate/nitrite response regulator protein NarL (NarL), and Ferrienterobactin receptor FepA (FepA) in sepsis.

Citrate forms stable complexes with Fe^2+^, enhancing its bioavailability and facilitating bacterial iron uptake through the ferric citrate transport system (FecABCDE) [64]. The significantly elevated citrate levels in urosepsis patients suggest a metabolic adaptation in response to iron homeostasis disruptions, particularly the host’s defense mechanisms against bacterial iron acquisition. While the host defends against infection by producing lipocalin-2 to sequester enterobactin, increased citrate may serve as an alternative iron source, bypassing siderophore reliance. Notably, 67% of *E. coli* strains—regardless of clinical group—carried the *fecA* gene, enabling ferric citrate uptake. This adaptation likely contributes to *E. coli*’s persistence in urosepsis. Additionally, citrate affects redox balance by accelerating Fe^2+^ oxidation and a process intensified during sepsis-related acidosis (lactic acidosis and metabolic acidosis) [64]. The interaction between acid-base disturbances and iron metabolism may thus play a key role in infection severity and pathogen adaptation.

The increased levels of (S)-malate in urosepsis patients may have significant implications for host metabolism and *E. coli* pathogenicity. Systemic alkalization induced by malic acid alters renal tubular reabsorption and citrate metabolism, leading to enhanced citrate excretion. This shift affects iron homeostasis, as citrate readily binds Fe^2+^ to form iron–citrate complexes, potentially reducing the necessity for bacterial siderophore-mediated iron acquisition. Consequently, the expression of siderophore systems, such as enterobactin and aerobactin, may be downregulated in response to the availability of iron–citrate complexes in the host environment. For *E. coli*, (S)-malate serves as a metabolically relevant alternative substrate, supporting energy production through succinyl-CoA synthetase activity in the TCA cycle [65]. Additionally, under anaerobic conditions, (S)-malate can be utilized via the fumarate, H^+^ symporter (DcuB), contributing to anaerobic respiration and energy conservation. This metabolic flexibility may enhance bacterial survival in fluctuating oxygen environments, such as those encountered in the urinary tract and bloodstream during urosepsis. The interplay between host-driven metabolic changes and bacterial adaptation mechanisms highlights the complexity of UPEC pathogenesis and suggests that malate availability could influence both iron acquisition strategies and energy metabolism in sepsis-causing *E. coli* strains.

Ubiquinone serves as a key component of the electron transport chain, facilitating aerobic respiration and maintaining membrane potential [66]. This is particularly relevant in the bladder, where oxygen availability fluctuates, influencing the expression of VFs. Oxygen-dependent ubiquinone synthase (UbiI) has been directly linked to the regulation of type I fimbriae expression, which plays a crucial role in *E. coli* adhesion to urothelial cells and subsequent colonization [67]. The findings from previous studies indicate that deletion of the *ubiI* gene impairs motility, reduces membrane potential, and disrupts fimbriae formation, further supporting the role of ubiquinone in bacterial virulence [67]. Given that biofilm formation is a major factor in chronic and recurrent UTIs, the ability of *E. coli* to utilize oxygen for ubiquinone synthesis may enhance its persistence in the urinary tract and increase its potential to cause severe infections, including urosepsis. In our study, the elevated ubiquinone levels in urosepsis patients highlight the metabolic adaptability of UPEC, reinforcing the link between aerobic respiration, type I fimbriae expression, and infection severity. This suggests that targeting ubiquinone biosynthesis could be a promising strategy for limiting *E. coli* pathogenicity and preventing severe UTI complications. In our opinion, elevated ubiquinone levels in patients with urosepsis are an indication of the high activity of UPEC strains, which contributes to increased expression of type I fimbriae.

Pectinesterase (*ybhC* gene) activity in *E. coli* may have significant implications for bacterial colonization and pathogenicity, particularly in the context of UTIs and urosepsis. The degradation of pectin by pectinesterase leads to the production of 2-dehydro-3-deoxy-D-gluconate, which may serve as a metabolic substrate [68]. Additionally, the resulting pectic acid may alter the local microenvironment, facilitating adhesion to epithelial cells and promoting infection progression [69,70].

The PhoP/PhoQ two-component system (OmpR family, response regulator PhoP) plays a crucial role in the adaptation of *E. coli* to hostile environments, including those encountered during UTIs and urosepsis [71]. By modulating gene expression in response to environmental stimuli such as low pH, antimicrobial peptides, and magnesium limitations, PhoP/PhoQ facilitates bacterial survival in the urinary tract and bloodstream. This regulatory system is particularly important for UPEC strains, which must withstand acidic urine and immune system defenses during infections. Furthermore, PhoP/PhoQ-mediated regulation extends beyond stress adaptation, influencing key metabolic pathways and VFs. The upregulation of genes associated with arginine and lysine metabolism may contribute to bacterial energy homeostasis and resistance to host-derived antimicrobial compounds. Additionally, PhoP/PhoQ influences outer membrane modifications, including lipid A remodeling, which enhances resistance to cationic antimicrobial peptides. Given its role in controlling invasiveness and persistence, the PhoP/PhoQ system represents a potential target for novel therapeutic interventions aimed at disrupting *E. coli* pathogenicity in UTIs and systemic infections.

The CpxA-CpxR two-component system (OmpR family, response regulator CpxR) is a crucial regulatory mechanism that enables *E. coli* to respond to envelope stress, a condition frequently encountered during UTIs and urosepsis. By modulating the expression of outer membrane porins, such as OmpF and OmpC, CpxR plays a role in controlling bacterial permeability to antimicrobial agents and environmental stressors, including variations in osmolarity and pH [72,73,74]. The differential regulation of these porins, particularly the suppression of OmpF and the upregulation of OmpC in response to high CpxR-P levels, contributes to the development of multidrug-resistant *E. coli* strains, which pose a significant challenge in clinical settings [75]. Beyond its role in envelope stress responses, the Cpx system is intricately linked to *E. coli* virulence. The increased expression of CpxR in urosepsis isolates suggests its involvement in enhancing bacterial fitness and persistence in the host. Notably, the regulation of P-type fimbriae, which are essential for adhesion to urothelial cells, is influenced by the Cpx signaling pathway [76,77,78]. This connection implies that CpxR not only facilitates bacterial survival under stressful conditions but also promotes colonization and biofilm formation in the urinary tract. Given its role in antibiotic resistance and virulence, targeting the Cpx regulatory system could be a potential strategy for combating persistent *E. coli* infections.

The NarL response regulator, a key component of the nitrate respiration regulatory network in *E. coli*, plays a crucial role in the adaptation of bacteria to anaerobic conditions, such as those encountered in the inflamed urinary tract during infection. By modulating the expression of nearly 100 genes, NarL facilitates bacterial survival and proliferation under low-oxygen conditions, which are commonly present in the bladder and kidneys during UTIs [79]. Additionally, nitrate respiration has been associated with enhanced bacterial fitness in the host, as it allows *E. coli* to sustain energy metabolism in oxygen-limited environments, thereby supporting persistent colonization. Beyond its metabolic function, NarL appears to be intricately linked to host–pathogen interactions, particularly in the context of inflammation. Research suggests that high levels of NarL expression can exacerbate mucosal immune responses, contributing to sustained inflammation and tissue damage. This heightened immune activation may create a feedback loop, promoting bacterial persistence and increasing the likelihood of recurrent UTIs. Furthermore, excessive inflammation driven by NarL-mediated mechanisms could escalate to systemic immune dysregulation, potentially leading to urosepsis. Given its dual role in bacterial survival and immune modulation, targeting the NarL regulatory pathway may represent a novel therapeutic approach to mitigate recurrent UTIs and prevent the progression to severe systemic infections.

The glutamate-aspartate transport system (GltL), including the ATP-binding protein GltL, plays a crucial role in the survival and pathogenicity of *E. coli* during UTIs and urosepsis. Glutamate transport is essential for bacterial adaptation to hostile environments, as it contributes to acid resistance, osmotic stress tolerance, and antibiotic resilience. The ability of UPEC to exploit host-derived glutamate enhances their metabolic flexibility, allowing them to persist within the urinary tract despite adverse conditions [80]. Moreover, the connection between glutamate transport and bacterial motility underscores its importance in *E. coli* virulence. The observed loss of the flagellar rotor protein FliG in *ΔgltS* mutants suggests that glutamate metabolism directly influences bacterial adhesion and invasion capabilities [80]. In our proteomic study, the overexpression of *gltL* in urosepsis strains compared to controls further supports the hypothesis that enhanced glutamate transport promotes bacterial colonization and persistence. This upregulation may facilitate biofilm formation and epithelial adherence, key factors in the pathogenesis of recurrent and severe infections. Given these findings, targeting glutamate transport pathways could offer a novel approach to mitigating *E. coli* virulence in UTIs and urosepsis.

Iron acquisition is a critical factor for *E. coli* survival and virulence, particularly in the context of systemic infections such as urosepsis. The outer membrane receptor FepA plays a central role in this process by mediating the uptake of ferric enterobactin, an iron-chelating siderophore essential for bacterial growth in iron-limited environments, such as the bloodstream and urinary tract [81]. In addition to its role in iron transport, FepA also serves as a receptor for colicins B and D, which may provide a competitive advantage by eliminating other bacterial strains. Regulation of *fepA* is controlled by the local repressor FepR; deletion or mutation of *fepR* leads to constitutive overexpression of *fepA*, a phenomenon associated with resistance to fluoroquinolone antibiotics such as norfloxacin and ciprofloxacin [82]. Moreover, several studies have reported upregulation of FepA in **septic environments**, supporting its relevance in clinical infections [83,84].

Our proteomic analysis revealed that while the *fepA* gene is present in both control and urosepsis groups, its expression is significantly upregulated in sepsis-causing *E. coli* strains. This suggests that FepA-mediated iron acquisition is particularly important during invasive infections, where iron availability is tightly restricted by host defense mechanisms. The selective expression of FepA in urosepsis isolates indicates a potential role in bacterial persistence and immune evasion, contributing to systemic dissemination and severe infection outcomes. Given its importance in pathogenicity, FepA represents a potential target for therapeutic strategies aimed at disrupting iron uptake in UPEC and preventing the progression from UTI to life-threatening sepsis. This opens the possibility of developing anti-FepA antibodies as a form of passive immunization or vaccine strategy against Gram-negative bacteria [85]. Siderofores are also being investigated as a target for siderophore–antibiotic conjugates (i.e., ‘Trojan horse’ antibiotics) [86,87] and may serve as a biomarker for pathogen detection or infection severity, especially in *E. coli* infections. Further research is warranted to assess FepA expression profiles in clinical sepsis samples, and to evaluate its potential incorporation into diagnostic panels or as a vaccine candidate for combating multidrug-resistant pathogens.

## 4. Materials and Methods

### 4.1. Selection of Strains for Genetic and Proteomic Studies

Sixty-four patients with UTIs caused by *E. coli* bacteria and symptoms of sepsis were selected to study in the Emergency Department (ED) of the Medical University of Gdansk (Poland) between 2019 and 2022. The control group consisted of 85 isolates from patients with UTIs without sepsis. The patients and isolates included in this study are shown in Appendix A. Our study included patients with uncomplicated urinary tract infections (UTIs) caused by *E. coli*, without structural abnormalities of the urinary tract or significant comorbidities such as diabetes, immunocompromised status, or active malignancy. Exclusion criteria also included recent urologic surgery, pregnancy or lactation, hospital-acquired infections, and antibiotic treatment within the previous two weeks.

Data from medical history, clinical examination, and results of the diagnostic tests for sepsis were obtained according to the Surviving Sepsis Campaign guidelines (https://www.sccm.org/clinical-resources/guidelines/guidelines/surviving-sepsis-guidelines-2021; 1 June 2025)

(https://machinelearningmastery.com/information-gain-and-mutual-information/; Critical Care Medicine: 4 October 2021). This study was approved by the local Bioethical Committee, Medical University of Gdańsk, Poland (NKBBN/133/2019), and each patient gave informed consent before being enrolled. SOFA (Sequential Organ Failure Assessment) scores equal to or greater than 2 were used as the cut-off value. Researchers did not have access to information that could identify individual participants during data collection.

*E. coli* cultures were processed by the respective clinical bacteriology laboratory and stored as frozen cultures (CryobankTM, MAST DIAGNOSTICA, GmbH, Reinfeld, Germany) before batch evaluations for bacterial characteristics.

Urosepsis was confirmed by genotyping isolates from the patients’ urine and blood using the PCR-Melting Profile method as previously described by Krawczyk et al. (2006) [88]. The same or similar fingerprint patterns (max. 1–3 bands difference) for isolates from the blood and urine suggested urosepsis. The description of the epidemic study results for the urosepsis strains under investigation was the subject of our previous work [59]. The absence of a predominant strain and high genetic diversity suggested that none of those strains were epidemic to the hospital.

### 4.2. Genetic Studies

Total DNA for genetic studies was extracted from a 1.5 mL culture by using a genomic DNA kit (EXTRACTME Genomic DNA KIT, BLIRT S.A., Gdansk, Poland) according to the manufacturer’s instruction.

#### 4.2.1. Phylogenetic Group Determination of *E. coli* Isolates

Phylogenetic groups were described using the method developed by Clermont, allowing the assignment of *E. coli* strains to one of eight phylogenetic groups (A, B1, B2, C, D, E, F, and clade I) [7,8]. OptiTaq PCR Master Mix (EURx Sp. z o.o., Gdańsk, Poland) was used for the quadruplex PCR.

#### 4.2.2. Detection of Virulence Gene by PCR

Twenty-nine genes with various functions were detected by simplex or multiplex PCR and PCR/RFLP for an additional seven genes included in the SPATE group, as follows: fimbrial and afimbrial adhesins—*fimH*, *mrkD*, *sfa*, *papG*, *afa/Dr*, *focG*, and *tosA* genes; invasin—*ibeA* gene; toxins: *usp*, *hly*, and *cnf1* genes; ATs, AIDA-I group—*aidA*, *ag43*, and *upaH* genes; SPATE group—*vat*, *pic*, *pic-like*, *pssA*, *boa*, *hbp*, and *tosB* genes; TAA group—*upaG* gene; capsule—*kspMTII* gene; iron uptake systems: siderophores and receptors—*iroN*, *iroB*, *entB*, *fepA*, *iutA*, *iucA*, *irp2*, *fyuA*, and *chuA* genes; and others—*fecA* and *iha* genes.

Condition reactions for the I multiplex PCR system (*papC*, *sfaD/E*, *cnf1*, *usp*, *fimG/H*, and *hlyA*) and the II multiplex PCR system (*kspMII*, *focG*, and *iha*) were as previously described by Adamus-Bialek et al. (2009) [89] and Krawczyk et al. (2020) [90]. Multiplex PCR for siderophore genes and their receptors (*entB*, *fepA*, *iucA*, *iutA*; *fyuA*, *irp2*, *iroN*, *and iroB*) were as described by Krawczyk et al. (2023) [59]. Multiplex PCR was also applied for two AT genes: *upa G* and *upaH.* Other genes were detected by individual, simplex PCR (*mrkD*, *afa/Dr*; *ag43*, *tosA*, *tosB*, *chuA*, *fecA*, *ibeA*, *aidaA-1*, and SPATE). Genes, primer sequences [36,51,88,89,90,91,92,93,94,95,96,97,98,99,100,101,102,103,104,105], and annealing temperatures for PCR are included in Appendix A.

Reference strains (CFT-073/ATCC 700928 and J96/ATCC 700336) and clinical *E. coli* isolates from the collection of the Gdansk University of Technology were used as positive controls.

Polymerase Taq (BIOLINE, Meridian Bioscience; BLIRT S.A., Gdansk, Poland, and EURX, Gdansk, Poland) was used for PCR. Electrophoretic detection was prepared in 1.2–1.5% agarose according to 0.5 × TBE buffer.

#### 4.2.3. PCR/RFLP for SPATE Genes

SPATE—*vat*, *pic*, *sat*, *pic-like* (U), *pssA*, *boa*, and *hbp*—were detected by PCR/RFLP (HaeIII) as previously described by Kotlowski et al. (2007) [103]. A PCR fragment of 618 bp indicates the presence of SPATE group ATs. The expected PCR/RFLP fragments are shown in Appendix A. Electrophoretic detection was prepared in 2% agarose in 0.5 × TBE buffer.

### 4.3. Proteomic Analysis

The clinical *E. coli* isolates (30 isolates from the control group with UTIs and 30 isolates from urosepsis) were cultured in artificial urine for proteomic analysis. To prepare the artificial urine, the following ingredients were used: lactic acid (0.1 g/L) (Sigma-Aldrich, Burlington, MA, USA), citric acid (0.4 g/L) (Sigma-Aldrich), sodium bicarbonate (2.1 g/L) (Sigma-Aldrich), urea (10 g/L) (Sigma-Aldrich), uric acid (0.07 g/L) (Sigma-Aldrich), creatinine (0.8 g/L) (Sigma-Aldrich), calcium chloride × 2 H_2_O (0.37 g/L) (Sigma-Aldrich), iron II sulphate × 7 H_2_O (0.0012 g/L) (Sigma-Aldrich), magnesium sulphate × 7 H_2_O (0.49 g/L) (Sigma-Aldrich), sodium sulphate × 10 H_2_O (3.3 g/L) (Sigma-Aldrich), di-potassium hydrogen phosphate (1.2 g/L) (Sigma-Aldrich), potassium dihydrogen phosphate (0.95 g/L) (Sigma-Aldrich), ammonium chloride (1.3 g/L) (Sigma-Aldrich), and sodium chloride (5.2 g/L) (Standard, Lublin, Poland). The pH 6,5 urine medium was achieved with hydrochloric acid. The whole was sterilized by filtration (VWR^®^, Radnor, PA, USA, Vacuum filtration, 150 mL, 0.2 µm PES filter Unit). Peptone and yeast extract were omitted from the medium to avoid interference in proteomic analyses.

A sterile liquid medium (10 mL) was inoculated with rejuvenated bacteria from solid cultures of *E. coli* strains. The culture was incubated for 24 h at 37 °C with continuous shaking at 70 rpm. After incubation, the culture was centrifuged at 4 °C for 10 min. Metabolism was then quickly quenched by adding cold methanol (1/20, *v*/*v*, −60 °C) to the supernatant. The solutions were pooled according to OD (600) measurements. For LC-MS analysis, only the supernatant was used to focus on extracellular proteins. We have prepared two samples: one pooled from UTI cultures and the second one from the urosepsis *E. coli* cultures. After the initial sample preparation, 5 µL of the protein solution was transferred to an Eppendorf tube. To this, 40 µL of 63 mM ammonium bicarbonate buffer, 2 µL of 2 mM dithiothreitol (DTT), 5 µL of 40 mM iodoacetamide, and 3 µL of trypsin solution were added. The tubes were incubated on a thermal shaker at 16 °C with shaking at 700 rpm for 16 h. After incubation, the samples were concentrated via vacuum evaporation for 3 h at 40 °C, and 50 µL of a 2% formic acid solution was added. Subsequently, 30 μL of each digested sample was transferred to LC vials with inserts and mixed for 5 min in a shaker. The final solutions were prepared for LC-MS analysis. The supernatant was lyophilized, and the resulting powder was resuspended in 50 μL of 0.1% formic acid. A 1 µL aliquot of the solution was injected into a Thermo Scientific UltiMate 3000 nano-LC system (Thermo Scientific, Waltham, MA, USA), coupled online with a SCIEX tripleTOF mass spectrometer (Sciex, Vaughan, ON, Canada). The trap cartridges (5 mm length, 300 μm ID), packed with C18 5 μm PepMap100 sorbent (Thermo Fisher Scientific, Waltham, MA, USA), were used to concentrate and desalinate the non-digested samples. A loading buffer (2% ACN, 0.1% TFA, H_2_O) was applied at a flow rate of 10 μL/min for 3 min. Peptides were separated on a 75 μm × 250 mm fused-silica analytical column packed with PepMap 2 μm sorbent (Thermo Fisher Scientific, Warszawa, Poland). The mobile phase gradient was a linear increase in phase B (0.1% FA in 80% ACN) from 2% to 40% over 20 min, followed by a transition to a 99% phase B over 40 min, with a flow rate of 300 nL/min. The column oven was set to 35 °C. Peptides eluting from the column were ionized using an OtpiFlow nano-ion source (ESI) and introduced into the SCIEX TT6600+ mass spectrometer in positive ionization mode. MS operation parameters included a spray voltage of +5.5 kV, nebulizer gas (N2) pressure of 14 psi, collision energy of +10 V, declustering potential of +90 V, and source temperature of 210 °C. The full scan range was set to m/z 350–1500, with fragmentation of the top 10 most intense precursor ions (charge +2 to +5). Collision energies ranged from 25 to 50 eV with a 10 eV spread.

Data collection was performed using Analyst v.1.8 (Sciex, Canada), and PeakView 2.2 (Sciex, Canada) was used for data visualization. ProteinPilot 5.0.2 software (Sciex, Canada) was employed to interpret the MS/MS spectra against the Uniprot database.

### 4.4. Metabolomic Analysis

LC-MS analysis was also conducted using a Thermo Fisher Scientific UltiMate 3000 nano-LC system (Thermo Fisher Scientific, USA), connected in-line to a TripleTOF mass spectrometer (Sciex, Framingham, MA, USA). The samples were pre-concentrated and desalted using trap cartridges, followed by a 3 min flow with a loading buffer (10 μL/min; 2% ACN in 0.1% FA). Metabolites were concentrated on a C18 5-μm PepMap100 sorbent (300 μm × 5 mm; Thermo Fisher Scientific, USA) and then transferred to a fused-silica analytical column packed with C18 5-μm PepMap100 sorbent (75 μm × 150 mm; Thermo Fisher Scientific, USA). The mobile phase gradient involved a linear increase in phase B (0.1% FA in 80% ACN) within phase A (0.1% FA in water), starting from 2% to 40% B over 20 min, and then ramping to 99% B for 40 min at a flow rate of 300 nL/min. The column oven was maintained at 35 °C. Metabolites eluting from the column were ionized using an Opti-flow nano-ion source (electrospray ionization) and introduced into the Sciex TT6600+ mass spectrometer, operated in both positive and negative ionization modes. MS parameters included a spray voltage of +5.5 kV (positive) and −5.0 kV (negative), nebulizer gas pressure (N2) at 14 psi, collision energy of +10 V/−10 V, declustering potential of +90/−90 V, and a source temperature of 180 °C. The full scan range was set from *m*/*z* 100 to 1500. Each full scan was followed by fragmentation of the top 10 most intense precursor ions with charges ranging from +1 to +2. Collision energy was set between 25 and 50 eV with a 10 eV spread. Data were collected using Analyst 1.8 (Sciex, Canada) software and PeakView 2.2, MasterView 2.1, and LibraryView 1.1 (Sciex, Canada), along with the NIST and Sciex All-in-One MS/MS spectra databases, which were used for data visualization and identification. To ensure high confidence in metabolite identification, stringent criteria were applied. Metabolites were annotated based on MS/MS spectral matches, with at least 50% of fragment peaks aligning with reference library spectra, a mass accuracy of less than 20 ppm, and matching isotopic patterns. Metabolites were identified at a confidence level of 2, supported by spectral similarity to library data and characteristic chromatographic behavior. The identification process was performed using two independent platforms, Agilent and Sciex, which allowed for cross-verification of results, leveraging the compatibility of the NIST and Sciex All-In-One libraries across both systems. Retention times of previously identified metabolites were incorporated to enhance annotation accuracy. By combining MS1 and MS/MS data obtained through different chromatographic methods, the workflow ensured robust and reliable metabolite identification, addressing potential methodological variations. False discovery rate (FDR) correction was applied using the MetaboAnalyst platform 5.0. Specifically, the Benjamini–Hochberg procedure was used to adjust the *p*-values and control the FDR at a set threshold (e.g., 0.05), minimizing false positives and improving the reliability and reproducibility of the results.

### 4.5. Statistical and Pathways Analysis

Fisher’s exact test was used to compare gene proportions. A result of *p* ≤ 0.05 was considered statistically significant. To calculate the exact Chi^2^, https://www.socscistatistics.com/tests/fisher/default2.aspx was used (accessed on 1 January 2025).

Orange Data Mining (v 3.36.1) was used for feature ranking based on information gain and decision tree algorithms [41].

Based on the metabolomic and proteomic data obtained from the clinical *E. coli* strains cultured in this study, we performed *t*-test analyses to identify the metabolites and proteins that differentiated the UTI and urosepsis groups. The metabolites used for this analysis were derived from the bacterial cultures of the clinical *E. coli* strains described in this study. These metabolites were crucial for selecting the metabolomic and proteomic data for statistical analysis and subsequent pathway selection. For the selected human metabolites and *E. coli* proteins, we assigned identifiers according to their KO numbers from the KEGG database: https://www.genome.jp/kegg/pathway.html (accessed on 10 June 2025)

We then mapped these metabolites and proteins into metabolic, proteomic, and enzymatic pathways in the KEGG database for *E. coli*. After analyzing these pathways and metabolites, we selected those that were specific and showed significant differences between the UTI and urosepsis cases. For the UTI group, we identified 799 proteins, and for the urosepsis group, 787 proteins, and for metabolites we focused on ~500 metabolites available in the Sciex AllinOne MS/MS metabolite database.

## 5. Conclusions

Our studies highlight key genetic, metabolic, and regulatory adaptations of *E. coli* that differentiate uncomplicated UTIs from urosepsis. Genes such as *sfa* and *upaG* emerged as significant markers distinguishing these clinical forms, with *UpaG* likely contributing to bacterial dissemination and bloodstream invasion. Iron acquisition pathways, particularly aerobactin and the Fec system, play a central role in virulence, especially under iron-restricted and oxidative stress conditions. Elevated citrate levels and metabolic shifts—such as increased ubiquinone production and enhanced glutamate utilization—further support bacterial persistence. Regulatory systems including CpxR, PhoP/PhoQ, and NarL enhance survival, immune evasion, and inflammation. These findings underscore the multifaceted strategies that *E. coli* employs to thrive in the urinary tract and bloodstream. Targeting iron metabolism, stress response pathways, and metabolic plasticity may offer novel therapeutic approaches to prevent the progression from UTIs to life-threatening urosepsis.

## Figures and Tables

**Figure 1 ijms-26-05681-f001:**
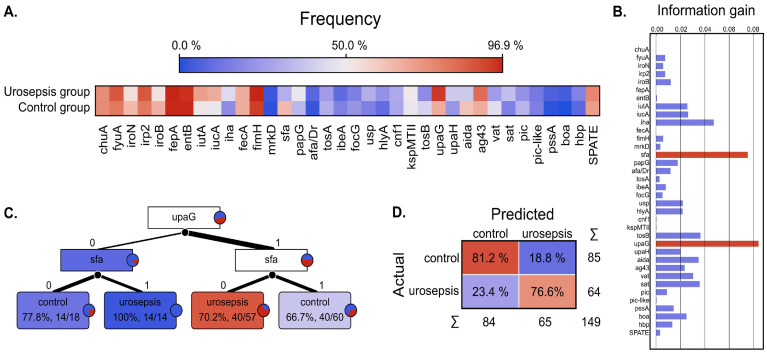
Binary classification results for urosepsis diagnostics. (**A**) the heatmap of the frequency of different virulence factors in the control and urosepsis groups. (**B**) Information gain analysis for feature selection; *sfa* adhesin encoding fimbriae S type and *upaG* gene encoding autotransporter; we paid special attention to autotransporters providing the highest score. (**C**) The classification tree for diagnostics of urosepsis. (**D**) Confusion matrix of the cross-validated classification model based on tree classifiers with a total classification accuracy of 79.2%; the percentage values represents the proportion of the actual target class. The heatmaps, feature ranking based on the information gain, and decision tree algorithm were performed using Orange Data Mining (v 3.36.1) [41]. Information gain provides information on data entropy reduction while splitting the data set into two target classes [42,43]. For the sake of analysis, those classes were the ‘control group’ and ‘urosepsis group’, respectively. The binary prediction model was validated using stratified 10-fold cross-validation. UpSet plots were generated using an R script with the UpSetR package (v 1.4.0).

**Figure 2 ijms-26-05681-f002:**
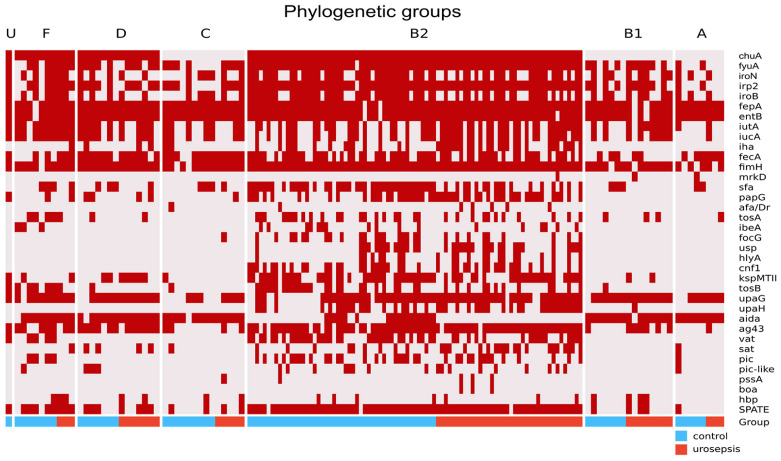
Heatmap of virulence factors detection in each phylogenetic group; red indicates detected virulence genes and the colored labels depict control (blue) and urosepsis (red) groups.

**Figure 3 ijms-26-05681-f003:**
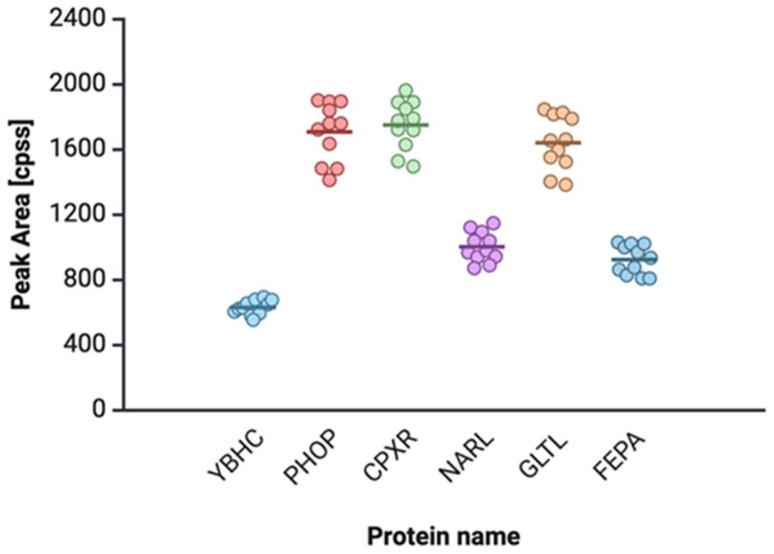
Proteins exclusively detected (presented as peak areas of respective peptides) in the proteomic profile of urosepsis-derived *E. coli* strains through untargeted LC-MS/MS analysis, absent in strains associated with uncomplicated urinary tract infections (UTIs). YBHC—pectinesterase; PHOP—phoP two-component system, OmpR family; CPXR—cpxR two-component system, OmpR family; NARL—key component of the nitrate respiration regulatory; GLTL—glutamate-aspartate transport system ATP-binding protein; FEPA—outer membrane receptor for ferric enterobactin.

**Figure 4 ijms-26-05681-f004:**
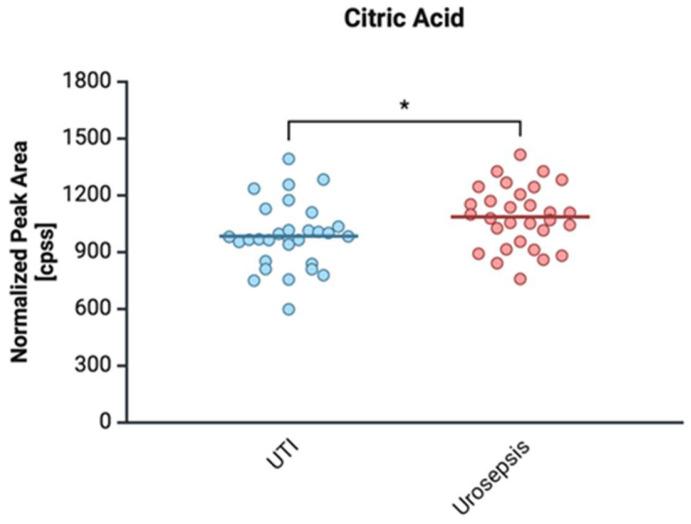
Comparison of citric acid levels in clinical *E. coli* isolates from patients with urinary tract infection (UTI) and urosepsis. Data are presented as mean ± standard deviation: UTI (984.6 ± 173.7) vs. urosepsis (1086.5 ± 162.5). * statistically significant levels *p* ≤ 0.05.

**Figure 5 ijms-26-05681-f005:**
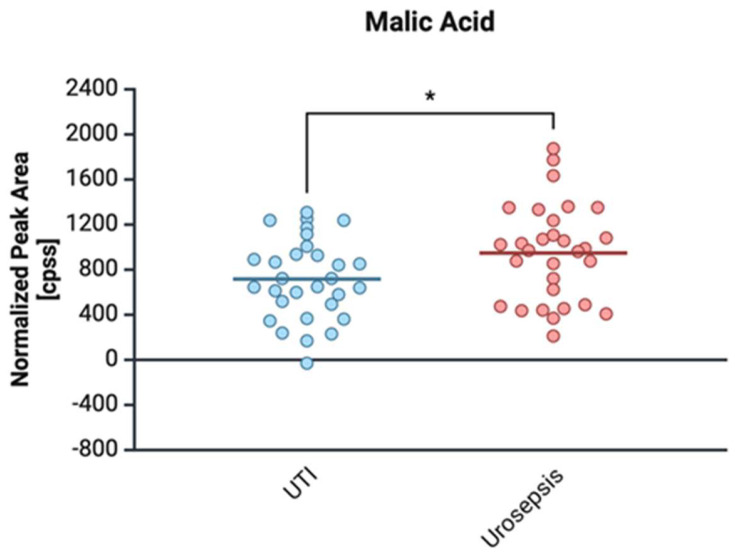
Comparison of malic acid levels in clinical *E. coli* isolates from patients with urinary tract infection (UTI) and urosepsis. Data are presented as mean ± standard deviation: UTI (716.67 ± 355.46) vs. urosepsis (947.75 ± 429.95); * statistically significant levels *p* ≤ 0.05.

**Figure 6 ijms-26-05681-f006:**
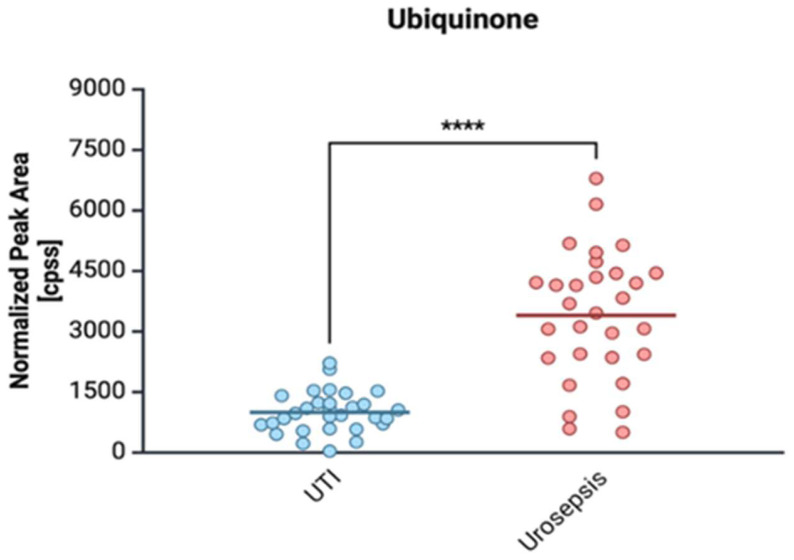
Comparison of ubiquinone levels in clinical *E. coli* isolates from patients with urinary tract infection (UTI) and urosepsis. Data are presented as mean ± standard deviation: UTI (998.18 ± 502.72) vs. urosepsis (3400.54 ± 1596.71); **** statistically significant levels *p* < 0.0001.

**Table 1 ijms-26-05681-t001:** Contribution of virulence factors—urosepsis vs. control groups. *p* *—*p* ≤ 0.05 was considered statistically significant.

Genes	Control Group n = 85 (%)	Urosepsis Group n = 64 (%)	*p* *
**Fimbrial adhesins**			
*fimG/H*	79/93%	62/97%	0.4669
*mrkD*	1/2%	2/3%	0.5771
*sfaD/E*	54/63%	20/31%	0.0001
*papG*	28/33%	31/48%	0.0641
*focG*	19/22%	10/16%	0.3676
**Afimbrial adhesins**			
*Afa/Dr*	2/2%	5/8%	0.1005
*tosA*	26/31%	16/25%	0.4312
**Invasin**			
*ibe*	14/16%	6/9%	0.2346
**Toxins**			
*usp*	14/16%	20/31%	0.0478
*hlyA*	8/9%	14/22%	0.0385
*cnf1*	24/28%	19/30%	0.8763
**Autotransporters**			
*tosB*	35/41%	13/20%	0.002
*upaG*	57/67%	60/94%	0.00001
*upaH*	20/24%	25/39%	0.033
*ag43*	57/67%	53/83%	0.014
*aidA*	52/61%	25/39%	0.003
Gene of SPATE group	60/71%	49/70%	1.00
*vat*	38/45%	16/25%	0.0161
*sat*	14/16%	23/36%	0.0077
*pic*	27/32%	14/22%	0.1514
*pic-like*	13/15%	9/14%	1.0
*pssA*	1/1%	4/6%	0.1184
*boa*	0/0%	3/5%	0.0594
*hbp*	7/8%	11/17%	0.1281
**Fe uptake**			
Enterobactin:			
*entB*	82/96%	61/95%	1.00
*fepA*	82/96%	62/97%	1.00
Salmochelin:			
*iroB*	56/66%	34/53%	0.13
*iroN*	54/64%	35/55%	0.31
Aerobactin:			
*iucA*	41/48%	43/67%	0.030
*iutA*	44/52%	45/70%	0.028
Yersiniabactin:			
*irp-2*	68/80%	56/88%	0.27
*fyuA*	68/80%	56/88%	0.27
Other:			
*chuA*	64/75%	48/75%	1.00
*fecA*	57/67%	43/67%	1.00
*iha*	17/20%	28/44%	0.002
**Capsule**			
*kspMTII*	41/48%	32/50%	1.0

## Data Availability

Data for this article are available at repository Most Wiedzy: https://mostwiedzy.pl/pl/open-research-data/virulence-factors,318023559359571-0. (1 June 2025). Data set: Wityk, P., & Krawczyk, B. (2025). Virulence factors (1–) [Dataset]. Gdańsk University of Technology. https://doi.org/10.34808/b4gf-7g37. (1 June 2025). The proteomic data, including protein names, the number of identified fragments, and the organism analyzed, as well as the genetic analysis data, such as gene names, gene presence, and expression levels, are available in the repository at the following link: https://doi.org/10.34808/p5e2-kp40 (1 June 2025). (clinical, genetic and proteomic) and https://doi.org/10.34808/s6at-g267 (1 June 2025) and https://doi.org/10.34808/nmrh-e263 (1 June 2025) (metabolomic). This repository ensures full transparency and accessibility of the research data for further analysis and validation.

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
