# Peer review of "Uropathogenic Escherichia coli Associated with Risk of Urosepsis—Genetic, Proteomic, and Metabolomic Studies"

_ijms, 2025, doi:10.3390/ijms26125681_

Round 1
Reviewer 1 Report
Comments and Suggestions for Authors
This paper addresses the important problem of optimal selection of tests to assess the potential occurrence of strains causing urosepsis, which is a significant clinical problem, in a patient. The authors described in detail in the introductory part the scientific background of the presented topic. Then, the results of the study were described, taking into account all possibilities of assessing factors related to the phylogenetic features of the pathogen. The description of the research methods and statistical tests used included individual stages of the study. Statistical methods are sufficient to assess the significance of the results. The discussion, like the introduction, is pervasive and could be shortened to the most important issues raised in the study. Similarly, the conclusions should be more concise, so that the reader can assess what the most important message of the work is. Tables and figures are correct and legible. Editorial note - in the introduction, in line 51, there is no explanation of the abbreviation UTIs (this explanation is found only later in the work). In general, the text is pervasive and could be slightly shortened to the most important theses and results.
Author Response
Reviewer #1:
Comments and Suggestions for Authors
- This paper addresses the important problem of optimal selection of tests to assess the potential occurrence of strains causing urosepsis, which is a significant clinical problem, in a patient. The authors described in detail in the introductory part the scientific background of the presented topic. Then, the results of the study were described, taking into account all possibilities of assessing factors related to the phylogenetic features of the pathogen. The description of the research methods and statistical tests used included individual stages of the study. Statistical methods are sufficient to assess the significance of the results. The discussion, like the introduction, is pervasive and could be shortened to the most important issues raised in the study. Similarly, the conclusions should be more concise, so that the reader can assess what the most important message of the work is. Tables and figures are correct and legible. In general, the text is pervasive and could be slightly shortened to the most important theses and results.
Response: We sincerely thank you for your insightful comments and the considerable time and effort dedicated to reviewing our manuscript. We fully agree that clarity is essential. In response, we have revised and reorganized the Introduction and Conclusion sections to reduce redundancy and enhance overall readability.
Although the virulence of Escherichia coli has been extensively addressed in the literature, the specific contribution of virulence and fitness factors in the context of urosepsis remains inadequately elucidated. Accordingly, our study aimed to provide a comprehensive characterization of these factors in relation to the risk of developing urosepsis. The Discussion section has been revised to more thoroughly explain the potential role of these factors in urosepsis pathogenesis, with selected passages condensed to improve clarity and conciseness. We trust that these revisions have significantly improved the manuscript’s clarity, coherence, and accessibility.
- Introduction: The following lines have been removed for clarity and to reduce redundancy: L46–48, L53–55, L93–95, L104–105, L262–264, and L311–316. Additionally, the wording of several sentences has been revised to improve clarity and coherence.
- Discussion: Lines L250–252 and L262–264 have been removed. Furthermore, selected passages have been revised and shortened to enhance conciseness and improve readability.
- Conclusion: Lines L684–730 have been removed. The revised conclusion (L670–682) has been substantially shortened to emphasize the key findings of the study.
- Editorial note - in the introduction, in line 51, there is no explanation of the abbreviation UTIs (this explanation is found only later in the work).
Response: Thank you for noticing this. It has been correctedin the Introduction issue (Line 31) abbreviation UTIs is explained: „Based on the Global Burden of Disease study, the number of urinary tract infec-tions (UTIs) increased by 66.45% between 1990 and 2021, affecting 4.49 billion people, with an age-standardized incidence rate (ASIR) of 5,531.88 per 100,000 [1].”
I would like to thank you for your thorough assessment of the manuscript and for all your substantive comments, which are extremely valuable to us.
Thank you for considering our revised manuscript for publication.
Sincerely,
Beata Krawczyk

Reviewer 2 Report
Comments and Suggestions for Authors
This manuscript presents a comprehensive multi-omics (genomics, proteomics, and metabolomics) analysis of Escherichia coli isolates from patients with urosepsis, compared with individuals with uncomplicated urinary tract infections. The authors employ powerful methodologies, including PCR genotyping, random proteomic analysis, metabolite profiling, and machine learning classification, to highlight distinct virulence patterns and potential biomarkers of sepsis risk. The manuscript is well-organized, methodologically sound, and offers valuable clinical implications. The supplementary file adds significant value, particularly with detailed peptide identifications, metabolite comparisons, and virulence factor cluster analysis. Although it requires several points of clarification and revision.
- Please provide details of any clinical or demographic matching between the sepsis group and the control group. Were there any confounding variables such as age, comorbidities, or antibiotic use?
- Please include a brief statement comparing the synthetic urine results to natural urine or referencing validation studies. This is particularly important for proteomic analysis and gene expression interpretation.
- Supplementary UpSet plots and HCA plots (Figures S4 and S5) reveal insightful clustering patterns. A brief mention of these results in the main Results/Discussion section would improve interpretation.
- S fimbriae variability: The unexpectedly high sfa ratio in control strains is noteworthy. Does this indicate a suppressive adaptation to bloodstream survival, or is it sample-specific variability?
- fepA overexpression: Its potential role as a therapeutic or diagnostic marker has not yet been developed. Consider a brief discussion or reference to studies linking fepA to iron acquisition in sepsis settings.
- Some legends (e.g., Figures S2-S5) may benefit from a broader context to facilitate their independent understanding.
- Please explicitly refer to Figures S4 and S5 in the main text where gene clustering and distribution are discussed.
- The language is generally clear, but professional editing would improve readability.
Author Response
Reviewer #2:
Comments and Suggestions for Authors
This manuscript presents a comprehensive multi-omics (genomics, proteomics, and metabolomics) analysis of Escherichia coli isolates from patients with urosepsis, compared with individuals with uncomplicated urinary tract infections. The authors employ powerful methodologies, including PCR genotyping, random proteomic analysis, metabolite profiling, and machine learning classification, to highlight distinct virulence patterns and potential biomarkers of sepsis risk. The manuscript is well-organized, methodologically sound, and offers valuable clinical implications. The supplementary file adds significant value, particularly with detailed peptide identifications, metabolite comparisons, and virulence factor cluster analysis. Although it requires several points of clarification and revision.
- Please provide details of any clinical or demographic matching between the sepsis group and the control group. Were there any confounding variables such as age, comorbidities, or antibiotic use?
Response: We thank the reviewer for his effort in reviewing our manuscript.
Following your suggestions, more information has been added. Information regarding the age and gender distribution of patients in each group is now provided in Table S3. The urosepsis group included 64 patients (32 women and 32 men), aged 24–80 years, while the control group comprised 85 patients with E. coli-associated UTIs without sepsis (63 women and 22 men), aged 20–81 years.
Also added clinical criteria for urosepsis patients: Body temperature < 36°C or > 38°C; Heart rate > 90 beats per minute; Respiratory rate > 20 breaths per minute or PaCO₂ < 32 mmHg; White blood cell (WBC) count < 4 × 10⁹/L or > 12 × 10⁹/L, with > 10% immature neutrophils.
In response to your suggestion, we have added the following clarification to the main text (Material and Methods section):
Line: 509-513 “Our study included patients with uncomplicated urinary tract infections (UTIs) caused by E. coli, without structural abnormalities of the urinary tract or significant comorbidities such as diabetes, immunocompromised status, or active malignancy. Exclusion criteria also included recent urologic surgery, pregnancy or lactation, hospital-acquired infections, and antibiotic treatment within the previous two weeks.”
- Please include a brief statement comparing the synthetic urine results to natural urine or referencing validation studies. This is particularly important for proteomic analysis and gene expression interpretation.
Response:
We are aware that artificial urine differs in composition from natural urine. We have compiled a comparison below in table:
Feature |
Artificial urine |
Natural human urine |
Main Components |
Phosphates (Na₂HPO₄, KH₂PO₄), NaCl, NH₄Cl, MgSO₄, glicerol |
Urea, creatinine, Na⁺, K⁺, Cl⁻, NH₄⁺, phosphates, sulfates, uric acid, and other metabolites |
Protein Content |
None |
Very low (<150 mg/day in healthy individuals) |
Organic Compounds |
Minimal – only glycerol |
High – urea, creatinine, uric acid, and various metabolites |
pH |
Manually adjusted, e.g., to 7.4 |
Variable: typically 4.5–8.0 (average ~6.0) |
Sterility |
Sterilized (autoclaving + 0.2 μm filtration) |
Not sterile – may contain bacteria, epithelial cells, etc. |
Osmolality |
Controlled by formulation (relatively low/moderate) |
Variable – usually 500–800 mOsm/kg, but can reach up to 1200 mOsm/kg |
Application |
Standardized medium for reproducible in vitro experiments |
Physiological matrix – used in biological models but with high variability |
Reproducibility |
High – precisely controlled composition |
Low – subject to inter-individual variability, diet, hydration, etc. |
We agree with Smith et al., 2018 and Zhang et al., 2020 „it does not fully replicate the complexity and variability of natural human urine, particularly about its organic content and protein composition. This limitation is especially relevant when interpreting results from proteomic analyses and gene expression studies, where the presence of endogenous proteins, peptides, and metabolites in natural urine can significantly influence biological responses. Therefore, where applicable, findings derived from synthetic urine should be validated against natural urine samples, as recommended in previous studies to ensure physiological relevance”
but: According us and other researchers synthetic urine provides a standardized and reproducible environment, eliminating the biological variability inherent in natural urine due to factors such as diet, hydration, and health status. Its defined composition enhances experimental consistency and enables precise interpretation of proteomic and gene expression data. Moreover, synthetic urine is sterile, safe to handle, and can be tailored to mimic specific urinary conditions, making it particularly useful in infection models, antimicrobial testing, and studies on biofilm formation.
Component |
Final Concentration (g/L) |
Supplier |
Lactic acid |
0.1 |
Sigma-Aldrich |
Citric acid |
0.4 |
Sigma-Aldrich |
Sodium bicarbonate |
2.1 |
Sigma-Aldrich |
Urea |
10.0 |
Sigma-Aldrich |
Uric acid |
0.07 |
Sigma-Aldrich |
Creatinine |
0.8 |
Sigma-Aldrich |
Calcium chloride × 2 H₂O |
0.37 |
Sigma-Aldrich |
Iron(II) sulfate × 7 H₂O |
0.0012 |
Sigma-Aldrich |
Magnesium sulfate × 7 H₂O |
0.49 |
Sigma-Aldrich |
Sodium sulfate × 10 H₂O |
3.3 |
Sigma-Aldrich |
Di-potassium hydrogen phosphate |
1.2 |
Sigma-Aldrich |
Potassium dihydrogen phosphate |
0.95 |
Sigma-Aldrich |
Ammonium chloride |
1.3 |
Sigma-Aldrich |
Sodium chloride |
5.2 |
Standard, Poland |
To prevent background interference in proteomic analyses, the medium was prepared without the addition of peptone and yeast extract.
In the Materials and Methods section, we found an error; instead of the composition of artificial urine, the composition of the M-9 medium was given. We have corrected it.
- Supplementary UpSet plots and HCA plots (Figures S4 and S5) reveal insightful clustering patterns. A brief mention of these results in the main Results/Discussion section would improve interpretation.
Response: There are only few reports addressing the role of autotransporters in urosepsis. Therefore, we find the co-occurrence of these findings particularly interesting. Therefore, we share your interest and have included them in the discussion.
A greater diversity in the co-occurrence of different types of autotransporters can be observed for the strains responsible for urosepsis.
The text has now been included in the „Results” section. (Line: 143-162)
„The most common combinations were upaG + aidA and upaG + aidA + ag43, both of which were observed in 9.4% of urosepsis strains, as well as upaG + aidA + ag43 + SPATE + hbp. Interestingly, the boa gene, which belongs to the SPATE family, was ex-clusively detected in urosepsis isolates and co-occurred with pssA in almost 5% of cases. By contrast, the control group exhibited the highest frequency of upaG + aidA (11.8%), with complex combinations of seven AT genes — including SPATE, upaG, ag43, vat, tosB, pic and either upaH or aidA — appearing less frequently. To evaluate patterns of gene co-occurrence within the autotransporter (AT) system, hierarchical clustering analysis (HCA) was also performed separately for the control group and the urosepsis group, using Jaccard distance and Ward linkage (Figure S4). In the control group, two major gene clusters were identified. The first cluster comprised upaG, aidA, tosB, pic, vat, agn43, and SPATE, indicating frequent co-occurrence of these genes in strains with lower virulence potential. A second, smaller cluster included upaH and sat, while genes such as pssA, hbp, and pic-like appeared more distantly grouped, suggesting limited presence or distinct functional roles. In contrast, the urosepsis group exhibited a more complex clustering pattern. A notably tight cluster involved boa and pssA—with boa being exclusively detected in urosepsis strains—suggesting its potential role in pathogenicity. Another extended cluster grouped aidA, hbp, pic, tosB, upaH, and vat, indicating coordinated expression of virulence-associated genes. Additionally, upaG, agn43, and SPATE formed a separate cluster, while sat and pic-like were grouped independently.”
In Discussion section: „Analysis of autotransporter (AT) gene co-occurrence revealed distinct patterns, with combinations including SPATE, ag43, and upaG more common in urosepsis isolates, suggesting their association with increased virulence. The boa gene, exclusive to urosepsis strains and often paired with pssA, may play a key pathogenic role. Hierarchical clustering showed simpler gene groupings in control strains—where upaG clustered with lower-virulence genes—while urosepsis strains exhibited more complex clusters, indicating coordinated expression of virulence factors”.
- S fimbriae variability: The unexpectedly high sfa ratio in control strains is noteworthy. Does this indicate a suppressive adaptation to bloodstream survival, or is it sample-specific variability?
Response: S fimbriae (sfa genes) are typically associated with uroepithelial and endothelial adhesion, The sfaDE genes are crucial for the formation and assembly of S fimbriae. I think that suppressive adaptation is not posssible because we didn’t detect sfa gene. The low ratio of sfa genes in urosepsis strains indicates that these bacteria do not have the ability to produce them. We can therefore consider the second version (sample-specific variability).
- fepA overexpression: Its potential role as a therapeutic or diagnostic marker has not yet been developed. Consider a brief discussion or reference to studies linking fepA to iron acquisition in sepsis settings.
Response: You're right to highlight the potential of FepA, a TonB-dependent outer membrane receptor involved in enterobactin-mediated iron uptake, as a point of interest in the context of sepsis. Although the role of FepA in iron acquisition is well established, its potential as a diagnostic or therapeutic marker remains underexplored. FepA is known to be overexpressed in response to iron starvation, enhancing bacterial growth and virulence. Interestingly, selective overexpression of FepA has been observed in urosepsis isolates, suggesting a role in bacterial persistence and immune evasion, which may contribute to systemic dissemination and severe infection outcomes. Regulation of fepA is controlled by the local repressor FepR; deletion or mutation of fepR leads to constitutive overexpression of fepA, a phenomenon associated with resistance to fluoroquinolone antibiotics such as norfloxacin and ciprofloxacin (Guérin et al., Overexpression of the Novel MATE Fluoroquinolone Efflux Pump FepA in Listeria monocytogenes Is Driven by Inactivation of Its Local Repressor FepR, PLoS One. 2014;9(9):e106340, doi:10.1371/journal.pone.0106340). Given its central role in pathogenicity, FepA represents a promising target for therapeutic strategies aimed at disrupting iron acquisition in UPEC and potentially preventing the progression from urinary tract infection (UTI) to life-threatening sepsis. Moreover, several studies (e.g., Biville F, Cwerman H, Létoffé S, Rossi MS, Drouet V, Ghigo JM, Wandersman C. Haemophore-mediated signalling in Serratia marcescens: a new mode of regulation for an extra cytoplasmic function (ECF) sigma factor involved in haem acquisition. Mol Microbiol. 2004 Aug;53(4):1267-77. doi: 10.1111/j.1365-2958.2004.04207.x. PMID: 15306027.; Cassat JE, Skaar EP. Iron in infection and immunity. Cell Host Microbe. 2013 May 15;13(5):509-519. doi: 10.1016/j.chom.2013.04.010. PMID: 23684303; PMCID: PMC3676888.) have reported upregulation of FepA in septic environments, supporting its relevance in clinical infections. Additionally, significant antibody recognition of ferric enterobactin-binding proteins like FepA could inhibit the spread of Enterobacteriaceae (Baghal et al., Production and immunogenicity of recombinant ferric enterobactin protein (FepA) International Journal of Infectious Diseases Volume 14, Supplement 3, September 2010, Pages e166-e170). This opens the possibility of developing anti-FepA antibodies as a form of passive immunization or vaccine strategy against Gram-negative bacteria. Siderofores (and FepA) are also being investigated as a target for siderophore–antibiotic conjugates (i.e., "Trojan horse" antibiotics) (Passari et al., Opportunities and challenges of microbial siderophores in the medical field. Appl Microbiol Biotechnol. 2023 Nov;107(22):6751-6759. doi: 10.1007/s00253-023-12742-7. Möllmann U, Heinisch L, Bauernfeind A, Köhler T, Ankel-Fuchs D. Siderophores as drug delivery agents: application of the "Trojan Horse" strategy. Biometals. 2009;22(4):615-624. doi:10.1007/s10534-009-9219-2) and may serve as a biomarker for pathogen detection or infection severity, especially in E. coli infections. Further research is warranted to assess FepA expression profiles in clinical sepsis samples, and to evaluate its potential incorporation into diagnostic panels or as a vaccine candidate for combating multidrug-resistant pathogens.
- Some legends (e.g., Figures S2-S5) may benefit from a broader context to facilitate their independent understanding.
Thank you for your valuable feedback, now is
Response: Thank you for your valuable feedback. Figure legends have been expanded with additional descriptions.
For greater convenience in comparing Figures S2 and S3 and for a consistent caption, I decided to create one S2 Figure with designations A and B.
Figure S2. Heatmap of E. coli isolates obtained from patients with urosepsis (A, left pane) and from control (urinary tract infection, UTI) cases (B, right pane), based on the distribution of virulence factor (VF) genes. The clustering was performed using the Average Linkage method with Spearman Rank Correlation as the distance metric (generated using Heatmapper: http://heatmapper.ca/expression/). Each row represents a clinical isolate, while each column corresponds to a specific virulence gene. The presence of a gene is indicated by a colored cell, while its absence is shown in darker colors. Hierarchical clustering allowed the identification of isolate groups with similar virulence profiles, suggesting potential pathotype-specific patterns among strains associated with urosepsis. The use of Spearman correlation enabled the analysis to focus on rank-based relationships between gene profiles, which is particularly suitable for binary presence/absence data and non-parametric distributions. The resulting dendrograms (on the left axes) reflect the degree of similarity among isolates and among virulence factors, revealing potential co-occurrence of gene sets and allowing inference of common virulence mechanisms. A greater diversity was observed among the control (UTI) strains, which clustered into five distinct groups, suggesting higher genetic variability in terms of virulence gene presence. In contrast, urosepsis isolates formed three main clusters, indicating a more homogeneous virulence profile (there seems to be more structured separation among clusters) that may reflect common pathogenic mechanisms associated with bloodstream infections. Inside each cluster, subgroups are included.
- Please explicitly refer to Figures S4 and S5 in the main text where gene clustering and distribution are discussed.
Response: The figures currently have changed numbers for Figure S3 and S4. This information is covered in Lines: 144 for Figure S3 and 165 for Figure S4 in Results section Lines: 332 and 335 in Discussion section.
I would like to thank you for your thorough assessment of the manuscript and for all your substantive comments, which are extremely valuable to us.
Thank you for considering our revised manuscript for publication.
Sincerely,
Beata Krawczyk
